# Discovery of an intermediate nematic state in a bilayer kagome metal ScV₆Sn₆

Camron Farhang[1], William R. Meier [2], Weihang Lu[1], Jiangxu Li [3], Yudong Wu[1], Shirin Mozaffari [2], Richa P. Madhogaria[2], Yang Zhang [3,4], David Mandrus [2,5] & Jing Xia [1] ✉

Nematicity, spontaneous breaking of rotational symmetry, is a ubiquitous phenomenon in correlated quantum matter. Here we show a phase transition in high-quality ScV₆Sn₆ bilayer kagome metal at a temperature $T^*$, occurring seven Kelvins below the charge density wave transition at $T_{CDW}$, as indicated by thermodynamic, transport, and optical measurements. This emerging intermediate phase does not exhibit spontaneous time-reversal-symmetry breaking, as evidenced by zero-field Sagnac interferometry. However, it displays a strong, spontaneous in-plane anisotropy between $T^*$ and $T_{CDW}$, revealed by transport and optical polarization rotation measurements. A pronounced depolarization effect detected by the Sagnac interferometer further confirms its nematic nature. Unlike AV₃Sb₅, this phase, alongside the recently discovered intra-unit cell nematic order at lower temperatures, presents a diverse landscape of nematicities at multiple length and temperature scales. Our findings highlight ScV₆Sn₆ as a prime candidate for realizing symmetry-breaking phases driven by charge density competition, kagome physics, and Van Hove singularities.

Spontaneous symmetry breaking underlines a multitude of fundamental natural phenomena, from the mass acquisition of elementary particles[1] to various solid-state matters in nature[2]. In the context of quantum materials, the breaking of lattice rotational symmetry, or nematicity, plays a crucial role in some of the most intriguing systems, including high-temperature cuprate superconductors[3], quantum Hall systems[4], and, more recently, layered kagome metals[5–9]. The vanadium kagome metal ScV₆Sn₆[10], with V atoms forming kagome bilayers (Fig. 1a), belongs to the extensive RT₆X₆ family of intermetallics (R = rare earth; T = V, Cr, Mn, Fe, Co; X = Ge, Sn). This family exhibits complex magnetism and nontrivial topological electronic properties[11–17]. Its band structure[18] features multiple Van Hove singularities (VHS) near the Fermi level, a Dirac point at the K point, and a flat band above the Fermi level $E_F$, suggesting the potential for novel correlated phases. Below $T_{CDW} \sim 80$ to 95K[10,18–23], ScV₆Sn₆ hosts a $\sqrt{3} \times \sqrt{3} \times 3$ CDW state tripling the unit cell along the c-axis[10], accompanied by a shift of VHS away from $E_F$[18]. Additionally, a recent STM study has revealed an intra-unit-cell nematic order and the associated Fermi surface deformation that occur at a much lower temperature of 70K[9]. Compared to the AV₃Sb₅ (A = K, Rb, Cs) family of kagome metals[24], which host both CDW[24] and nematic orders[5–8], ScV₆Sn₆ exhibits unique CDW instabilities due to the smallness of Sc atoms[25,26], giving rise to competing CDW fluctuations[19,20,27]. Understanding if these instabilities lead to new CDW-driven nematicities[6] is of key importance.

Notably, below $T_{CDW}$, a pronounced second resistivity drop occurs at $T^* \approx 86K$, as shown in Fig. 1b. This feature has been clearly observed only in ScV₆Sn₆ crystals with similar growth conditions[18], while hints of it appear as a long tail in temperature-dependent resistivity in other reports[19,23]. The nature of this feature at $T^*$ has not been previously discussed to the best of our knowledge.

[1]Department of Physics and Astronomy, University of California, Irvine, Irvine, CA 92697, USA. [2]Department of Materials Sciences and Engineering, University of Tennessee-Knoxville, Knoxville, TN 37996, USA. [3]Department of Physics and Astronomy, University of Tennessee-Knoxville, Knoxville, TN 37996, USA. [4]Min H. Kao Department of Electrical Engineering and Computer Science, University of Tennessee-Knoxville, Knoxville, TN 37996, USA. [5]Materials Science and Technology Division, Oak Ridge National Laboratory, Oak Ridge, TN 37831, USA. ✉e-mail: xia.jing@uci.edu

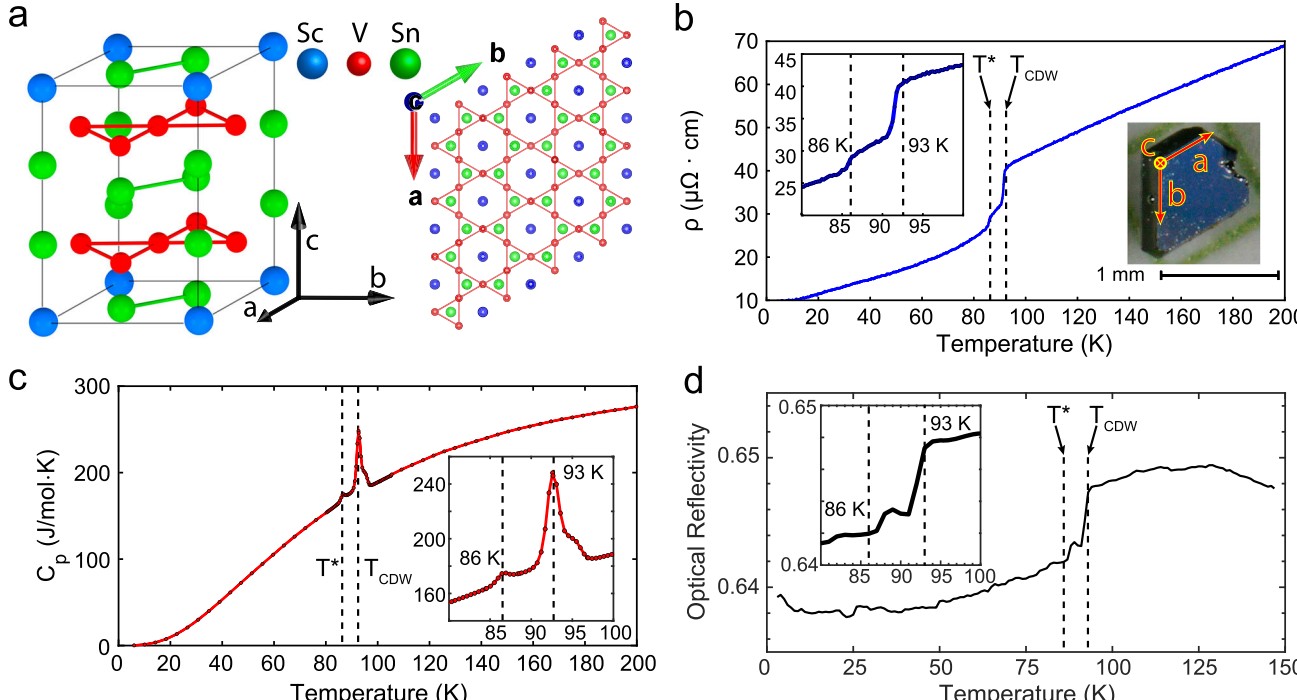

**Fig. 1 | Emergence of an intermediate phase between T\* and T_CDW inside the CDW state of ScV₆Sn₆.** a ScV₆Sn₆ structure generated using VESTA (left), with the Kagome lattice in the ab-plane (right). b ab-plane resistivity ρ showing sharp drops when the sample is cooled through $T_{CDW}$ and $T^*$. Left inset: resistivity near transition temperatures. Right inset: image of the measured ScV₆Sn₆ single crystal sample. c Specific heat capacity $C_p$ exhibits a major peak at T_CDW and a minor peak at $T^*$, serving as direct evidence for double phase transitions. Inset: $C_p$ near transition temperatures. Dots are individual data points. d Optical reflectivity measured with the 1.55 μm optical beam focused onto a single spot on the ab-plane surface. The sharp changes at both $T_{CDW}$ and $T^*$ demonstrate the intrinsic nature of the "double transition". Reflectivity measurements at various spots are presented in Supplementary Fig. 1. Inset: reflectivity near transition temperatures.

Here we report transport, thermodynamics, optical reflectivity, polarization rotation, and Sagnac interferometry measurements in high-quality ScV₆Sn₆ crystals, revealing an unusual nematic phase emerging in the narrow temperature range between $T_{CDW}$ and $T^*$. Upon further cooling below $T^*$, in-plane rotational symmetry is restored as competing charge-density wave fluctuations subside, distinguishing this intermediate nematic state from any previously reported nematic phase.

## Results

First, we report an intermediate phase inside the CDW state in high-quality ScV₆Sn₆ samples synthesized using the tin flux method[10]. They are nicely faceted metallic hexagonal blocks about 1mm in size with optically flat ab-plane surfaces (Fig. 1b inset). The ab-plane resistivity ρ during cooldown is plotted in Fig. 1b. The CDW transition is evident as a sharp drop in in-plane resistivity on cooling through $T_{CDW} \approx 93K$, consistent with previous reports[10,18–23]. Below $T_{CDW}$, a pronounced second resistivity drop at $T^* \approx 86K$ suggests an additional phase transition. The room-temperature resistivity of ScV₆Sn₆ samples used in this study is 90μΩ · cm, yielding a residual resistivity ratio (RRR) of 9, comparable to ref. 18 but about twice as large as other reports[10,18–23]. The improved RRR suggests higher crystal quality and may explain why the T\* transition is only clearly observed in this work and in ref. 18.

The drop in resistivity alone doesn't guarantee a phase transition. A thermodynamic signature, often considered a "smoking gun", has never been reported for a second phase transition in ScV₆Sn₆[10,18–23]. To elucidate whether $T^*$ marks a true phase transition, we have performed specific heat $C_p$ measurements on the same sample during cooldown, as shown in Fig. 1c. A sharp main peak in $C_p$ centered at 93K signals the expected CDW transition[10,19,22]. Additionally, a peak in $C_p$ emerges at 86K, coinciding with the lower temperature ab-plane resistivity jump, providing a smoking gun for a phase transition at $T^*$.

It is crucial to rule out the trivial scenario in which the $T^*$ transition arises merely from a sample region with a reduced $T_{CDW}$. To address this, we performed optical reflectivity measurements at 1.55μm wavelength during cooldown at numerous focused spots spread over the ab-plane surface. One such measurement is presented in Fig. 1d: the optical reflectivity drops from 0.648 to 0.644 when the sample is cooled below $T_{CDW}$, and it drops further from 0.644 to 0.642 at T\*. Similar reflectivity drops at both $T_{CDW}$ and $T^*$ were observed at other spots (Supplementary Fig. 1), confirming the intrinsic nature of the "double transition". We observe that $T^*$ varies by ∼ 1K across different locations, while $T_{CDW}$ remains uniform, suggesting that $T^*$ marks a more delicate phase transition, more sensitive to local environments than the CDW transition. In a metal that is isotropic in the ab-plane, a reduced optical reflectivity usually indicates a decreased plasma frequency and hence less numerous carriers, consistent with what was found in Hall effect measurements on a similar sample[18]. The increased carrier mobility[18] due to a decreased electronic scattering by CDW fluctuations[20] accounts for the lower resistivity in the CDW phase, despite diminished density of states.

Next, we test spontaneous time-reversal symmetry breaking (TRSB). Beyond the lattice-translational symmetry breaking characteristic of the CDW phase, spontaneous TRSB in ScV₆Sn₆ has been reported as hidden magnetism in a muon spin relaxation (μSR) study[22]. To investigate whether TRSB is responsible for the T\* transition, we employ a zero-area Sagnac interferometer[28] that exclusively detects magneto-optic Kerr effect (MOKE) signals arising from microscopic TRSB and rejects any non-TRSB effects such as anisotropy[29–31]. This is because the sourcing aperture for one light is also the receiving aperture for the other time-reversed counterpropagating light. Consequently, Onsager's relations guarantee zero signal in the absence of microscopic TRSB. Spatial imaging and temperature-dependent measurements of spontaneous MOKE signals $\theta_K$ were performed on the ab-

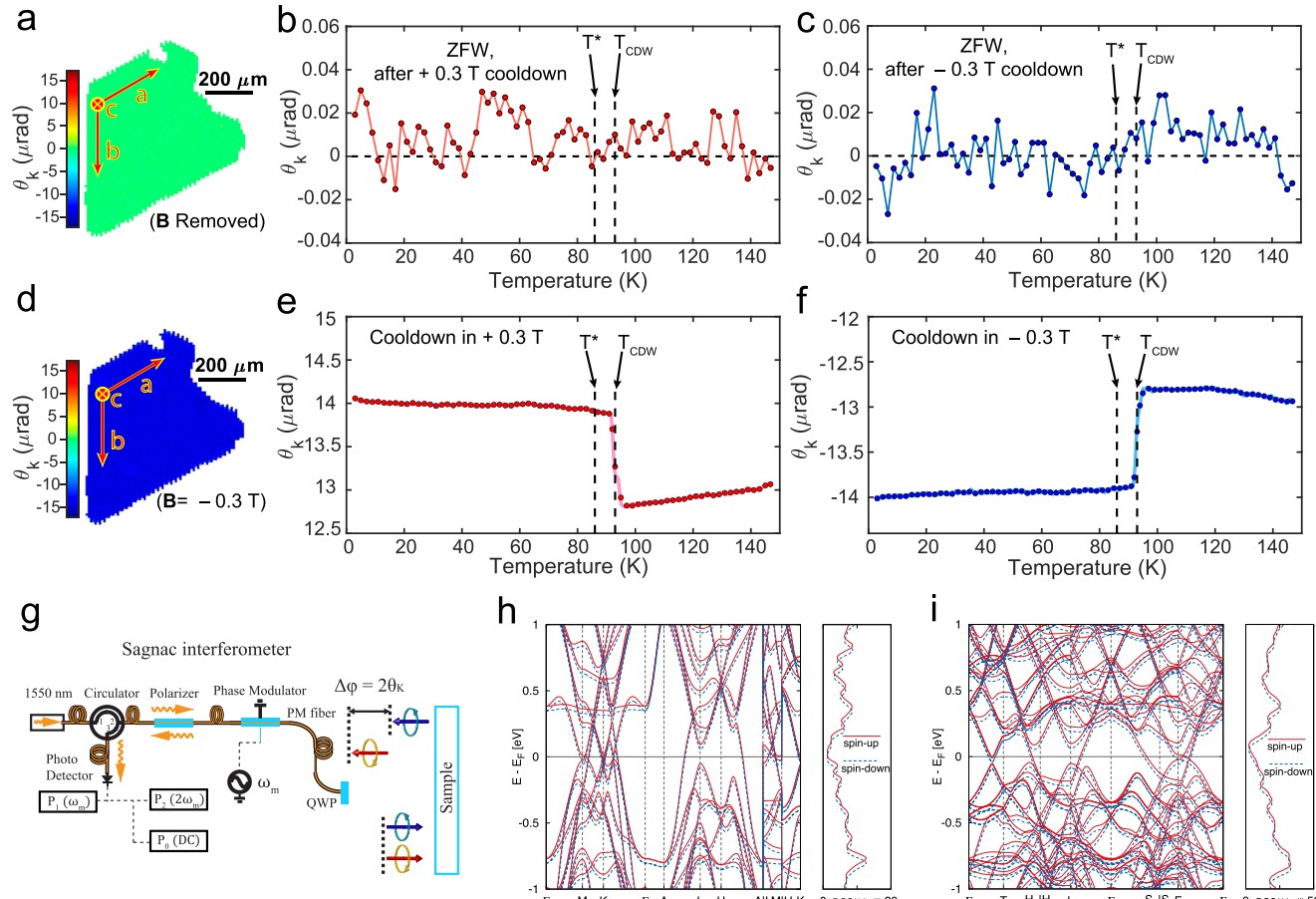

**Fig. 2 | MOKE θ_K in ScV₆Sn₆ and absence of spontaneous time-reversal symmetry breaking (TRSB).** MOKE signal $\theta_K$ is measured using a Sagnac interferometer microscope, first during cooldown recording $\theta_K$ in a magnetic field, then during zero field warmup (ZFW) with the field removed to record spontaneous $\theta_K$. See Methods for details. **a** Image of spontaneous $\theta_K$ at 3K after the removal of a **B** = − 0.3T magnetic field, showing zero signal across the sample. **b**, **c** ZFW after + 0.3T and −0.3T field cooldowns show no discernable onset of spontaneous $\theta_K$ at either $T_{CDW}$ or $T^*$, with ±10nrad uncertainty. Together these spontaneous $\theta_K$ traces and image show no evidence for TRSB. More measurements of spontaneous $\theta_K$ are presented in Supplementary Fig. 3. **d** Image of $\theta_K$ at 3K in **B** = − 0.3T showing uniform $\theta_K$ across the sample due to Pauli paramagnetism. **e**, **f** $\theta_K$ during + 0.3T and −0.3T field cooldowns showing a paramagnetic MOKE response with a

sharp increase below $T_{CDW}$, but no change across $T^*$, indicating the $T^*$ transition is not coupled to the magnetic degree of freedom. More measurements of $\theta_K$ in magnetic fields are presented in Supplementary Fig. 2. **g** Schematics of a zero-area-loop fiber-optic interferometer that is only sensitive to TRSB (MOKE $\theta_K$) effects, which is independent of α. The fiber-optic head can be scanned to simultaneously acquire reflection and MOKE images. **h**, **i** Calculated band structure and density of states of ScV₆Sn₆ under an external magnetic field **B** for **h** the high-temperature phase and **f** the CDW phase. The spin-up and spin-down channels are marked as the red and blue lines, respectively. The spin polarization at the Fermi level is doubled in the CDW phase, accounting for an increase in $\theta_K$ despite reduced density of states.

plane surface using a microscope version of the Sagnac interferometer[32–35]. To align the chiralities of possible TRSB domains, a training magnetic field is applied during cooldown and subsequently removed at the lowest temperature. Figure 2a) depicts a spontaneous $\theta_K$ image acquired at $T = 3K$, taken after removing a 0.3T magnetic field during cooldown. No discernable spontaneous MOKE signal was observed in the image. To further search for any changes in the signal across $T^*$ with the highest resolution, we perform zero field warmups (ZFW) at numerous locations with different training fields. One example of ZFW after **B** = + 0.3T training is shown in Fig. 2b, and another example after **B** = − 0.3T training is presented in Fig. 2c. Despite these efforts, we didn't observe any onset of $\theta_K$ at either $T_{CDW}$ or $T^*$ with an uncertainty of ± 0.01μrad. In fact, across the whole temperature range, there is no discernable onset at any temperature with an uncertainty of ± 0.02μrad. Additional ZFW measurements are presented in Supplementary Fig. 3, further reinforcing the conclusion that there is no evidence for TRSB at any temperature in our sample, which stands in stark contrast to the μSR study[22]. It is noteworthy that compared to our ScV₆Sn₆ crystal, the CDW transition in the μSR study

occurs at a significantly lower temperature of 80K[22] compared to most other studies[10,18–23], suggesting a presence of impurities that may account for the observed hidden magnetism in their sample.

In the absence of spontaneous TRSB in our sample, the MOKE signal in a magnetic field originates from Pauli paramagnetism[10,18] in ScV₆Sn₆, which lacks strongly magnetic atoms. Figure 2d presents a MOKE image obtained at $T = 3K$ in a magnetic field of −0.3T, revealing a small but exceptionally uniform $\theta_K$ signal of −14.0μrad with a spatial variance of only 0.1μrad across the entire ab-plane surface. Temperature-dependent $\theta_K$ measurements (Fig. 2e and Supplementary Fig. 2a for **B** = + 0.3T cooldowns, Fig. 2f and Supplementary Fig. 2b for **B** = − 0.3T cooldowns) reveal a 10% increase in $\theta_K$ at $T_{CDW}$. While this initially appears to contradict the concurrent reduction in carrier density[18], calculations of spin split bands under a magnetic field revealed a significant enhancement of spin polarization in the CDW phase, leading to larger MOKE signals. Figure 2h, i show the calculated band structure and density of states of ScV₆Sn₆ under an external magnetic field **B** for the high-temperature and CDW phases, respectively, with spin-up and spin-down channels marked in red and blue,

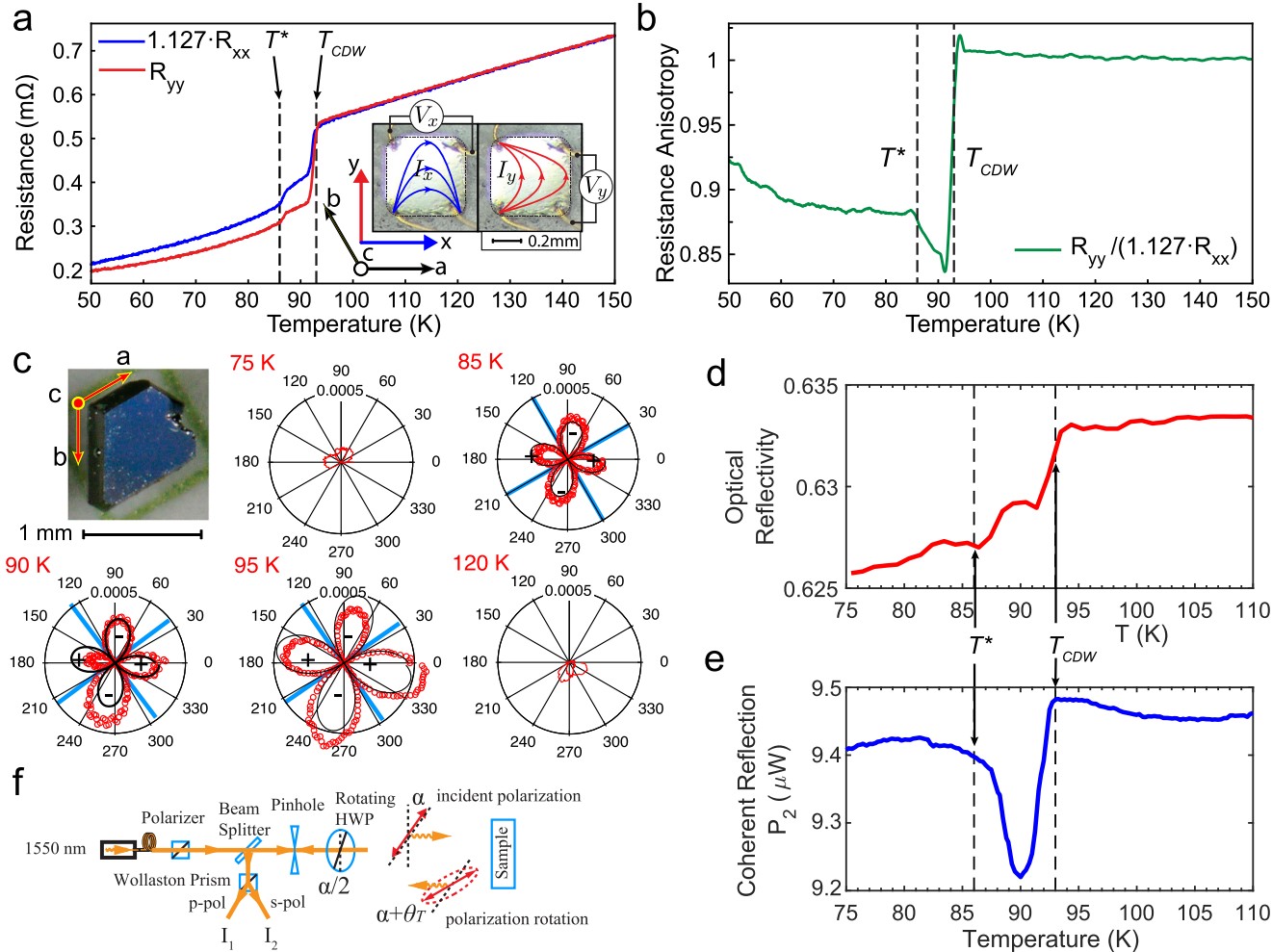

**Fig. 3 | Broken ab-plane rotational symmetry. a** Resistances $R_{xx} \equiv \frac{V_x}{I_x}$ and $R_{yy} \equiv \frac{V_y}{I_y}$ of a ScV$_6$Sn$_6$ sample polished into a square plate. $1.127 R_{xx}$ is used in the plot to account for a slight deviation from a perfect square-shape and misalignment of electrical contacts. $1.127 R_{xx}$ and $R_{yy}$ trace each other almost perfectly when cooling down from 150K, but develop a pronounced anisotropy below $T_{CDW}$. Such anisotropy decreases but doesn't fully disappear when further cooling down below $T^*$. Inset: pictures of the square sample with illustrations of a-b and x-y axes as well as wire configurations. Curved arrows represent the expected current paths for $I_x$ and $I_y$ in the ab-plane. **b** Temperature trace of $\frac{R_{yy}}{1.127 R_{xx}}$ as a phenomenological parameter quantifying the ab-plane transport anisotropy. The largest anisotropy occurs between $T_{CDW}$ or $T^*$. **c**, Polar plots of optical polarization rotation measured at one location on the ab-plane and at various temperatures. The development of a four-leaf clover pattern with alternating "+" and "-" signs between 95K and 85K indicates intrinsic ab-plane anisotropy. The full scale of the polar plots is 0.0005rad. The crossed blue lines represent fitted principal birefringent axes, which are close to the a-axis and its normal direction. Inset: image of the measured ScV$_6$Sn$_6$ single crystal with crystal axes labeled. Polarization rotation measurements at various locations are presented in Supplementary Fig. 4. **d**, **e** Depolarization effects revealed by comparing optical reflectivity (red) and the coherent reflection $P_2$ measured in the Sagnac interferometer (blue). $P_2$ measures the part of reflected optical power that remains coherent after reflection in a Sagnac interferometer, where depolarization effects in the sample, such as anisotropy and chirality, reduce such coherence. See Methods for details. Profoundly, cooling from $T_{CDW}$ to $T^*$, the reflectivity (red) shows a drop-plateau-drop while $P_2$ exhibits a single pronounced dip, revealing a large depolarization effect between $T_{CDW}$ and $T^*$, which is consistent with the observed transport and optical anisotropy. Cooling further below $T^*$, $P_2$ recovers from the dip and starts gradually to follow the trend of reflectivity, signaling the weakening and possible disappearance of the ab-plane anisotropy. $P_2$ measurements at various locations are presented in Supplementary Fig. 5. **f**, Schematics of polarization rotation setup based on a Wollaston prism that measures polarization rotation $\theta_T$ as a function of incident polarization $\alpha$, which is achieved by rotating a half-wave plate (HWP) by $\frac{\alpha}{2}$.

respectively. In the high-temperature phase, spin polarization at the Fermi level is 0.10/cell, increasing to 0.19/cell in the CDW phase, explaining the $\theta_K$ enhancement despite the reduced density of states.

A more intriguing observation is the absence of any change in $\theta_K$ across $T^*$, echoing the lack of any discernible kink in the magnetic susceptibility $\chi$ at $T^*$ in a similar crystal[18]. Since Pauli paramagnetism arises from the imbalance in spin-polarized carriers, this decoupling between the $T^*$ transition and the magnetic channel implies that the $T^*$ transition is likely associated to a rearrangement of charge carriers rather than a change in their total quantity.

Nematicity[4] in the intermediate state is a possible form of charge rearrangement that spontaneously breaks rotational symmetry. In ScV$_6$Sn$_6$, a recent STM study revealed an intra-unit-cell nematic order

accompanied by Fermi surface deformation[9]. The intra-unit-cell nematic order emerges at a lower temperature (70K) and in a different atomic layer than the CDW order, suggesting a weaker correlation with the CDW order[9]. In contrast, our directional resistance and optical polarization rotation measurements below reveal a distinct type of nematic order in ScV$_6$Sn$_6$ that is strongly linked to $T_{CDW}$ and $T^*$.

Zero-magnetic-field in-plane transport anisotropy is revealed in directional transport, as shown in Fig. 3a. A second ScV$_6$Sn$_6$ crystal was carefully cut into a 500μm square in the ab-plane with 200μm thickness along the c-axis, as shown in the inset of Fig. 3a. Four 25μm-diameter gold wires were connected to the four corners to measure resistances $R_{xx} \equiv \frac{V_x}{I_x}$ and $R_{yy} \equiv \frac{V_y}{I_y}$, where the in-plane x and y directions are defined as along and perpendicular to the a-axis of the kagome

lattice, respectively. Above $T_{CDW}$, $R_{xx}$ and $R_{yy}$ closely track each other, with a small offset (factor of 1.127) attributed to minor misalignments of electrical contacts. Therefore, we plot $1.127R_{xx}$ and $R_{yy}$ in Fig. 3a, showing near-perfect overlap when cooling down from 150K to $T_{CDW}$. However, below $T_{CDW}$, a pronounced anisotropy develops, which diminishes but doesn't completely vanish below $T^*$. To better visualize the temperature dependence of the in-plane transport anisotropy, we define $\frac{R_{yy}}{1.127R_{xx}}$ as a phenomenological anisotropy parameter and plot it in Fig. 3b. Above $T_{CDW}$, this ratio remains within $1 \pm 0.01$, confirming rotational symmetry in the kagome plane. At $T_{CDW}$, a sudden anisotropy suddenly appears as $\frac{R_{yy}}{1.127R_{xx}}$ drops to 0.84 within a fraction of a Kelvin. Upon further cooling past $T^*$, the ratio recovers to 0.90, signaling a rapid but incomplete recovery of rotational symmetry.

To rule out sample inhomogeneity as the source of macroscopic transport anisotropy, we performed optical polarization rotation measurements, which also serve to determine the anisotropic principal axes[7,32,36]. Optical rotation ($\theta_T$) at 1.55 μm wavelength was measured under normal incidence onto the ab-plane while varying the incident polarization angle α. Figure 3c presents polar plots of $\theta_T(\alpha)$ at various temperatures, with the inset depicting the sample and crystallographic axes. Between $\sim 85$K and $\sim 95$K, $\theta_T$ exhibits both sign and magnitude variations, reaching $\pm 0.5$mrad in a two-fold symmetry pattern described by $\theta_T = \theta_P \sin(2\alpha - A)$. This clearly indicates spontaneous breaking of the six-fold in-plane rotational symmetry. A fitted parameter $A \sim 30°$ suggests that the principal anisotropy axis (blue lines in Fig. 3c) aligns with the a-axis of the kagome lattice. Above $T_{CDW}$, $\theta_T$ is negligible, indicating preserved rotational symmetry. Below $T^*$, its behavior becomes more complex, displaying significant spatial variations. Supplementary Fig. 4 presents $\theta_T$ measured at multiple locations: at point R, $\theta_T$ diminishes below $T^*$, whereas at point M and L, the two-fold symmetry pattern persists down to 70K and 60K, respectively. This inhomogeneity may explain why resistance anisotropy decreases but doesn't fully disappear below $T^*$.

Intrinsic nematicity also manifests itself in the depolarization effects detected by the Sagnac interferometer. As described in Methods, linear and circular birefringence can cause depolarization to the reflected circularly polarized light beams. Consequently, in a zero-area Sagnac interferometer, some reflected light loses coherence and does not participate in the interference, appearing as a drop in the coherent reflection $P_2$ (and $P_0$ and $P_1$). Such dips are clearly visible between $T_{CDW}$ and $T^*$ in $P_2$ (and $P_0$) traces presented in Supplementary Fig. 5, indicating strong depolarization effects in this temperature range. In contrast, the MOKE signal $\theta_K$ is calculated as the ratio between $P_1$ and $P_2$, and is not influenced by depolarization effects. To ascertain that the observed dip in $P_2$ is not due to a change in optical reflectivity, we compare optical reflectivity (red, Fig. 3d) and the coherent reflection $P_2$ (blue, Fig. 3e). Notably, these two signals are uncorrelated between $T_{CDW}$ and $T^*$ indicating strong depolarization effects, which is evident in Fig. 4a where we plot optical depolarization (red line) as the coherent reflection $P_2$ (Fig. 3e) divided by reflectivity (Fig. 3d) normalized to 80K. Similar to polarization rotation, spatial variations are observed in the magnitude of the $P_2$ dip, ranging from 3% to 1% at various locations (Supplementary Fig. 5).

## Discussion

In Fig. 4a, we summarize the nematic orders in ScV$_6$Sn$_6$ by plotting the scaled anisotropy observed in various local measurements. The anisotropic polarization rotation (yellow) represents fitted $\theta_P$ values from Fig. 3c using $\theta_T = \theta_P \sin(2\alpha - A)$, while the optical depolarization (red) is calculated as the coherent reflection $P_2$ (Fig. 3e) divided by reflectivity (Fig. 3d), both highlighting the presence of the intermediate anisotropic phase between $T^*$ and $T_{CDW}$ in high quality crystals. The intermediate phase is well separated in temperature from the recently discovered intra-cell nematic order below 70K in a STM study[9],

represented in Fig. 4a by the anisotropic Bragg peaks (blue) calculated as the ratio between Bragg peak intensities at $\mathbf{q}_3$ and $\mathbf{q}_2$ (adopted from ref. 9). Unlike clear signatures observed at $T^*$ for the intermediate phase, no anomalies were detected in transport, optical, or specific heat measurements for the intra-cell nematic order. This suggests that the latter is a much more subtle phenomenon, likely of a different origin.

These findings parallel the existence of two distinct temperatures associated with two-fold anisotropy in the CDW state of single-layer kagome metals AV$_3$Sb$_5$[24]. A broad range of experiments indicate the breaking of the underlying $C_6$ rotational symmetry[5-7,37,38] either at $T_{CDW}$ or at a much lower temperature $T'$. Initial nematic susceptibility measurements in CsV$_3$Sb$_5$ reported a diverging anisotropic (E$_{2g}$) elastoresistivity response below $T^*$[6], serving as evidence for electronic nematicity, that is, the partial melting of CDW that restores translational symmetry but breaks rotational symmetry[4]. However, more recent careful experiments[39,40] on CsV$_3$Sb$_5$ reported the absence of the E$_{2g}$ signal, suggesting that the $T'$ transition is not nematic in nature. Even more intriguing, a recent transport study[41] on CsV$_3$Sb$_5$ found that while unperturbed samples show no transport anisotropy, the application of either a magnetic field or strain induced a pronounced transport anisotropy below $T'$. These complex and seemingly contradictory findings suggest the coexistence of two distinct nematic states below $T'$ in AV$_3$Sb$_5$, making it difficult to disentangle them. In contrast, the clear separation between intra-cell nematic below 70K[9] and the intermediate nematic state in ScV$_6$Sn$_6$ allows easier independent study, which is further facilitated in the intermediate phase by the transport, optical, and thermodynamic signatures without any external magnetic field or strain. Future nematic susceptibility measurements[39] will be crucial in determining whether these two states arise from electronic nematic instability.

Another likely origin of the unusual intermediate nematic phase in ScV$_6$Sn$_6$ is the unique competition between different CDW instabilities[19,20,27]. Typically, materials tend to break symmetry upon cooling, resulting in states of lower symmetry. However, counter examples exist, such as the quantum Hall stripe to isotropic phase transition from 100 mK to 20 mK at half-filled high Landau levels restoring in-plane rotational symmetry[42], and the ferromagnetic to antiferromagnetic transition during cooling in intermetallic magnetic compound FeRh restoring time-reversal symmetry[43-45]. In ScV$_6$Sn$_6$ varied CDW phases have been predicted[25] and the lattice degrees of freedom have been shown experimentally to play a fundamental role in establishing the $\sqrt{3} \times \sqrt{3} \times 3$ CDW ground state[19,20,27]. In a nesting-driven CDW arising from Peierls instability, the diverging electronic susceptibility and phonon-softening occur at the same wavevector $\mathbf{q}$, and cause a static CDW at $\mathbf{q}_s$. In a typical electron-phonon-coupling (EPC) driven CDW[46], the strength of EPC peaks at $\mathbf{q}_{EPC}$ in the vicinity of phonon-softening regions, resulting in a static CDW at $\mathbf{q}_{EPC}$. Conversely, in ScV$_6$Sn$_6$ inelastic X-ray scattering found that EPC peaks at $\mathbf{q}_s = (\frac{1}{3}, \frac{1}{3}, \frac{1}{3})$ corresponding to the observed static $\sqrt{3} \times \sqrt{3} \times 3$ CDW, while the softest phonon modes occur at $\mathbf{q}^* = (\frac{1}{3}, \frac{1}{3}, \frac{1}{2})$ corresponding to a competing short-range $\sqrt{3} \times \sqrt{3} \times 2$ CDW[20]. We note that none of the diffraction measurements[20,27,47] have detected evidence of a long-range $\sqrt{3} \times \sqrt{3} \times 2$ CDW order, which could potentially break rotational symmetry. The dominance of the $\mathbf{q}^*$ mode at higher temperatures is surpassed by the $\mathbf{q}_s$ mode below $T_{CDW}$, leading to the long-range $\mathbf{q}_s$ CDW state. The proximity of $T^*$ to $T_{CDW}$ suggests that the observed nematicity is closely linked to the lattice degrees of freedom. It is plausible that remnants of the competing $\mathbf{q}^*$ mode slightly below $T_{CDW}$ further lower the rotational symmetry, giving rise to the observed nematicity. At a lower temperature, as the competing $\mathbf{q}^*$ mode subsides, rotational symmetry is restored. In our ScV$_6$Sn$_6$ sample this process must take place at a well-defined temperature $T^*$ in a sizable fraction of the bulk volume giving rise to the observed signatures in

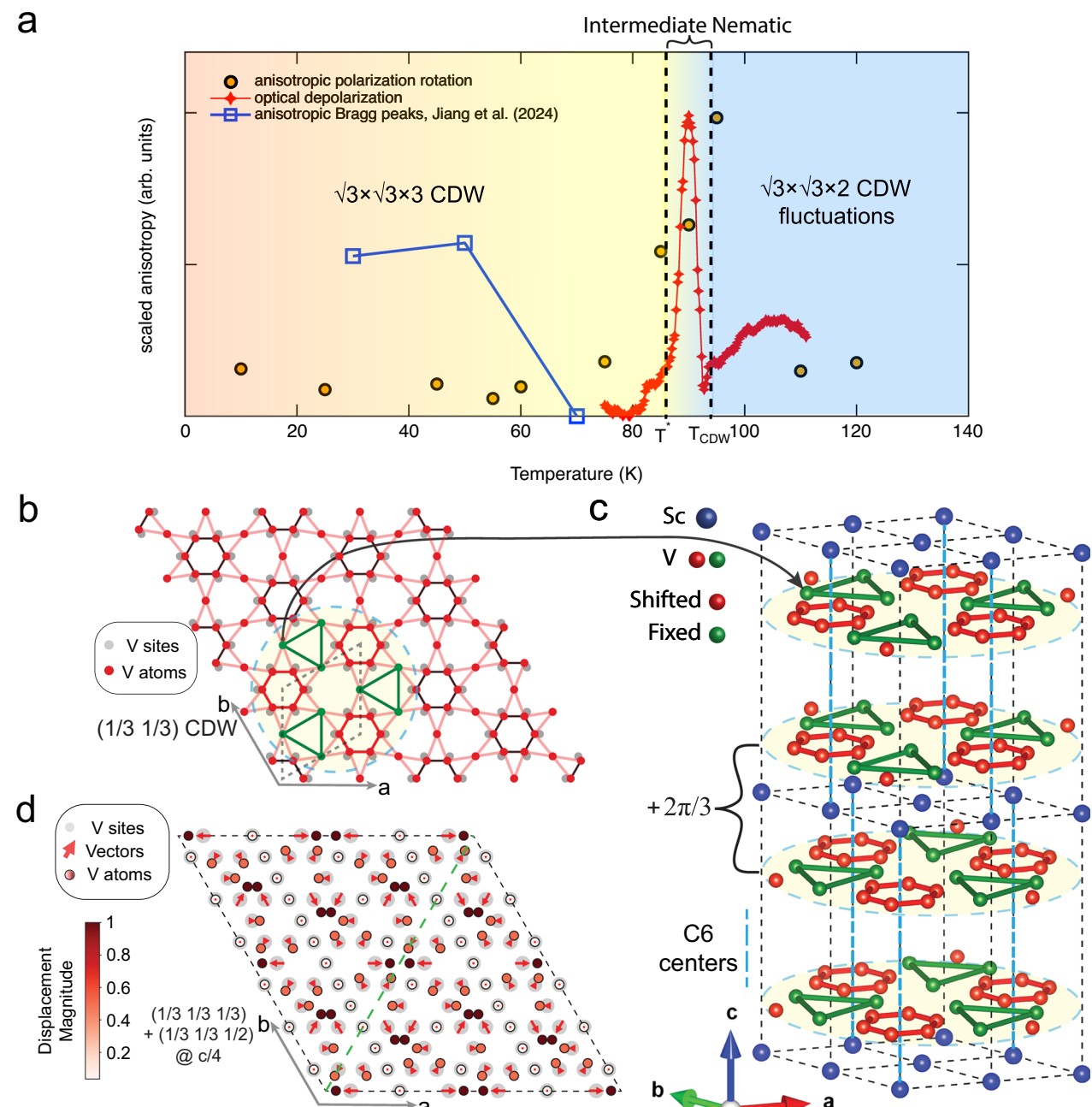

**Fig. 4 | Summary of nematic phases in ScV$_6$Sn$_6$ and potential sources of the intermediate nematic phase. a** Diagram of previously reported intra-unit cell nematic order below 70$K$[9] and the intermediate nematic state reported here. Anisotropic polarization rotation (yellow) represents fitted $\theta_P$ values from Fig. 3c using $\theta_T = \theta_P \sin(2\alpha - A)$, and optical depolarization (red) is calculated as the coherent reflection $P_2$ (Fig. 3e) divided by reflectivity (Fig. 3d), both indicating an intermediate anisotropic phase between $T_{CDW}$ and $T^*$ where $\sqrt{3} \times \sqrt{3} \times 3$ and $\sqrt{3} \times \sqrt{3} \times 2$ CDWs compete. Anisotropic Bragg peaks (blue) is calculated as the ratio between Bragg peak intensities at $\mathbf{q_3}$ and $\mathbf{q_2}$ adopted from a recent STM study[9] suggesting an intra-unit cell nematic order below 70$K$. **b** Illustration of the kagome lattice in the CDW phase with an exaggerated in-plane $\sqrt{3} \times \sqrt{3}$ CDW modulation for Vanadium atoms with wave vector $(\frac{1}{3}, \frac{1}{3})$, and 3 times larger in-plane unit cell (dashed black lines). Within the dashed blue circle, displaced and fixed atoms are colored red and green. **c** Stacking of CDW states depicted in **b** with $\frac{2\pi}{3}$ interlayer phase shift breaks rotational symmetry. The accumulated $\frac{2\pi}{3}$ phase difference between the top and bottom layer results in breaking of the $C_6$ symmetry due to a misalignment of the $C_6$ centers. **d** Mixing of competing $\sqrt{3} \times \sqrt{3} \times 3$ CDW with $\mathbf{q_s} = (\frac{1}{3}, \frac{1}{3}, \frac{1}{3})$ and $\sqrt{3} \times \sqrt{3} \times 2$ CDW with $\mathbf{q}^* = (\frac{1}{3}, \frac{1}{3}, \frac{1}{2})$, plotted at height $z = c/4$. The result is an anisotropic modulated lattice marked by the dashed green line. Original vanadium lattice sites of the kagome lattice are marked by gray circles, and displaced vanadium atoms are marked by circles with black edges with inside colors ranging from white (zero displacement) to red (maximum displacement). The darkest red on the color bar is normalized to the largest displacement magnitude. Red arrows at each gray circle represent the displacement vectors from the original lattice site to the displaced atom.

specific heat and resistance, while it persists below $T^*$ at other locations as detected in local optical anisotropy measurements (Supplementary Fig. 4), resulting in some remnant resistance anisotropy below $T^*$ (Fig. 3b).

One possible scenario considers the stacking degree of freedom. Below $T_{CDW}$, the static $\mathbf{q_s}$ CDW state preserves in-plane C$_6$ rotational symmetry within each kagome layer, as illustrated in Fig. 4b, where only Vanadium atoms are shown for clarity, though Sc and Sn atoms

also undergo shifts. However, in the presence of the competing $q^*$ CDW, perfect interlayer phase matching may be softened, allowing a finite interlayer phase shift. In the case of $CsV_3Sb_5$ recent Raman measurements[48] have proposed a $\pi$ phase interlayer shift being responsible for the observed $C_6$ symmetry breaking in the $q = (\frac{1}{2}, \frac{1}{2}, \frac{1}{2})$ CDW. For the $q_s = (\frac{1}{3}, \frac{1}{3}, \frac{1}{3})$ CDW in bilayer kagome metal $ScV_6Sn_6$, this would correspond to an inter-bilayer $\frac{2\pi}{3}$ phase shift. Figure 4c illustrates such a staggered configuration where $C_6$ symmetry is broken due to a misalignment of the $C_6$ centers, which may be restored at lower temperatures once the $q^*$ mode is suppressed to enforce perfectly stacked layers. This stacking would locally reduce rotational symmetry from six-fold to three-fold, but preserve six-fold symmetry on average across layers. In contrast, the $\pi$ phase shift in the proposed $2 \times 2 \times 2$ CDW phase[48] of $AV_3Sb_5$ break symmetry globally to two-fold, not being restored by averaging. Thus a $2\pi/3$ phase shift is unlikely to account for the rotational symmetry breaking we observe.

Alternatively, the mixing of $q_s$ and $q^*$ CDWs could directly lead to anisotropy. Although $q_s$ and $q^*$ share the same in-plane component $(\frac{1}{3}, \frac{1}{3})$, their relative phase varies across kagome layers resulting in anisotropy in some layers. For example, in Fig. 4d, we plot the lattice distortion due to mixing of these two CDWs at height $z = 0.25c$. A relative phase shift of $\Delta\phi = \pi/12$ results in an anisotropic state marked by the dashed green line. The $C_6$ symmetry is restored at lower temperatures when the $q^*$ mode diminishes.

In summary, $ScV_6Sn_6$ presents a diverse landscape of two distinct and well-separated nematicities at different length and temperature scales. This is distinct from the nematic state reported in kagome superconductors $AV_3Sb_5$[5–8]. Our results highlight $ScV_6Sn_6$ as a fertile platform for realizing symmetry-breaking phases arising from a unique combination of competing CDW instabilities, kagome physics, and Van Hove singularities.

## Methods
### Crysal growth
$ScV_6Sn_6$ samples were synthesized using the tin flux method, starting with an atomic ratio of Sc/Lu:V:Sn = 1:6:60. Sc/Lu pieces (Alfa Aesar, 99.9%), V pieces (Alfa Aesar, 99.8%), and Sn shot (Alfa Aesar, 99.9999%) were placed into an alumina Canfield crucible set. The crucible assembly was sealed in a fused silica ampule, heated to 1150 °C over 12 hours, maintained at this temperature for 15 hours, and then cooled to 780 °C over 300 hours. At 780 °C, the flux was separated from the crystals by inverting the ampule and centrifuging. This process produced hexagonal metallic crystals with typical lateral dimensions ranging from 0.5 to 4 mm. The elemental composition and approximate stoichiometry of the structure were characterized and confirmed via scanning electron microscopy (SEM), energy-dispersive spectroscopy (EDS), and powder x-ray diffraction (XRD). The detailed results of these characterizations are presented in refs. 10,18.

**Specific Heat** measurements were carried out using the heat capacity option of the Physical Property Measurement System (PPMS) by Quantum Design with the crystal mounted with Apiezon N grease. Specific heat data were collected by PPMS during cooldown by applying heat pulses and analyzing the subsequent cooling trace.

### Electrical Transport
Sample resistivity and anisotropic resistances were measured using a resistance bridge. Electric contacts were made with $25\,\mu m$-diameter gold wires that are connected to the four corners of the samples (Fig. 3a inset) using a home-made micro-spot-welding setup. This results in a small contact area ($< 50\,\mu m$) and low contact resistance ($< 0.1\,m\Omega$).

**Optical Reflectivity** is measured in normal reflection of a $1.55\,\mu m$ wavelength optical beam that is focused onto a single spot on the ab-plane surface. The recorded reflected optical power is compared with that from a reference sample (a highly reflective mirror) to obtain the optical reflectivity.

**Sagnac MOKE measurements** are performed using a zero-loop fiber-optic Sagnac interferometer as shown in Fig. 2g. The beam of light is routed by a fiber circulator to a fiber polarizer. After the polarizer the polarization of the beam is at 45° to the axis of a fiber-coupled electro-optic modulator (EOM), which generates 4.6 MHz time-varying phase shifts $\phi_m \sin(\omega t)$, where the amplitude $\phi_m = 0.92\,rad$ between the two orthogonal polarizations that are then launched into the fast and slow axes of a polarization maintaining (PM) single-mode fiber. Upon exiting the fiber, the two orthogonally polarized linearly polarized beams are converted into right- and left-circularly polarizations by a quarter-wave plate (QWP) and are then focused onto the sample. After reflection from the sample, the same QWP converts the reflected beams back into linear polarizations with exchanged polarization axes. The two beams then pass through the PM fiber and EOM but with exchanged polarization modes in the fiber and the EOM. At this point, the two beams have gone through the same path but in opposite directions, except for a phase difference of $\Delta\varphi$ from reflection off the magnetic sample and another time-varying phase difference by the modulation of EOM. This nonreciprocal phase shift $\Delta\varphi$ between the two counter-propagating circularly polarized beams upon reflection from the sample is twice the Kerr rotation $\Delta\varphi = 2\theta_K$. The two beams are once again combined at the detector and interfere to produce an optical signal $P(t)$:

$$P(t) = \frac{1}{2}P\left[1 + \cos\left(\Delta\varphi + \phi_m \sin(\omega t)\right)\right] \tag{1}$$

where $P$ is the returned power if the modulation by the EOM is turned off. For MOKE signals that are slower than the 4.6 MHz modulation frequency used in this experiment, we can treat $\Delta\varphi$ as a slowly time-varying quantity. And $P(t)$ can be further expanded into Fourier series with the first few orders listed below:

$$\begin{aligned} P(t)/P = \frac{1}{2}&[1 + J_0(2\phi_m)] \\ &+ (\sin(\Delta\varphi)J_1(2\phi_m))\sin(\omega t) \\ &+ (\cos(\Delta\varphi)J_2(2\phi_m))\cos(2\omega t) \\ &+ 2J_3(2\phi_m))\sin(3\omega t) \\ &+ \dots \end{aligned} \tag{2}$$

where $J_1(2\phi_m)$ and $J_2(2\phi_m)$ are Bessel J-functions. Lock-in detection was used to measure the first three Fourier components: the average (DC) power ($P_0$), the first harmonics ($P_1$), and the second harmonics ($P_2$). And the Kerr rotation can then be extracted using the following formula:

$$\theta_K = \frac{1}{2}\Delta\varphi = \frac{1}{2}\tan^{-1}\left[\frac{J_2(2\phi_m)P_1}{J_1(2\phi_m)P_2}\right] \tag{3}$$

### Sagnac measurement of depolarization effects
The above calculations of the Sagnac interferometer assume the absence of any depolarization effects in the sample, i.e., after the right- and left-circularly polarized waves are reflected from the sample surface, they will remain in perfect circular polarizations. In the presence of linear or circular birefringence and dichroism of the sample, however, these depolarization effects will not guarantee

preserving perfect circular polarizations. And the two reflected beams, after passing the quarter-wave plate again, become elliptical instead of being perfectly linearly polarized, causing the leakage of some optical power into the "wrong" polarization axes of the optical fiber. And a small fraction of the light will thus be incoherent with the major beams and won't participate in the interference. The remaining coherent parts of optical powers need to be multiplied by a correction factor of $(1 - D)$, where D is a small number quantifying the depolarization effects from the sample. For example, the second harmonics $P_2$ (denoted as coherent reflection in the main text and in Fig. 3e) is changed from $(\cos(\Delta\varphi)J_2(2\phi_m))\cos(2\omega t)P$ to $(1 - D)(\cos(\Delta\varphi)J_2(2\phi_m))\cos(2\omega t)P$, showing up as a "drop" in $P_2$. Since a decrease in optical reflectivity can cause a drop in the reflected total power P and hence a drop in $P_2$ even without any depolarization effects, it is necessary to compare measured coherent reflection $P_2$ and optical reflectivity to determine whether the observed "drop" in $P_2$ is indeed caused by depolarization. We note that the MOKE signal ($\theta_K = \frac{1}{2}\tan^{-1}\left[\frac{J_2(2\phi_m)P_1}{J_1(2\phi_m)P_2}\right]$ is calculated using the ratio of $P_1$ and $P_2$, both of which experience the same reduction in the presence of depolarization effects. Therefore, the depolarization effects will not affect the $\theta_K$ measurement.

### Polarization rotation

As shown in Fig. 3f, the beam of light is routed through a free-space polarizer to produce a linearly polarized beam. A polarization-independent beam splitter (half mirror) transmits half of the beam and reflects the other half, which is discarded. The transmitted beam passes through a pinhole and then a half-wave plate (HWP), which is mechanically rotated such that its principal fast axis is at an angle $\frac{\alpha}{2}$ to the polarization direction of the beam. The resulting beam after the HWP has its polarization direction rotated by angle $\alpha$. The beam then passes through the optical window of the cryostat and is reflected by the sample. The returned light beam passing the same pinhole a second time is, in general, elliptical with the major axis rotated by the total polarization rotation $\theta_T + \alpha$. After passing through the HWP a second time, its polarization direction is rotated by $-\alpha$, and becomes $\theta_T$. And the same polarization-independent beam splitter reflects half the returned beam towards a Wollaston prism, which separates and directs two orthogonal polarizations s and p toward two balanced detectors. The recorded intensities of s and p-polarization components are $I1$ and $I2$, respectively. The Wollaston prism is rotated at a $\frac{\pi}{4}$ angle such that with a gold mirror calibration piece $I1$ and $I2$ are "balanced": $\Delta I = I1 - I2 = 0$. With polarization rotation $\theta_T$ from the sample, an imbalance $I1 - I2$ emerges:

$$\frac{I1 - I2}{I1 + I2} = \sin(2\theta_T) \qquad (4)$$

hence:

$$\theta_T = \frac{1}{2}\arcsin\left(\frac{I1 - I2}{I1 + I2}\right) \qquad (5)$$

### Density functional theory (DFT) Calculations

All ab initio calculations were performed using the Vienna Ab-initio Simulation Package (VASP) based on density functional theory (DFT). The exchange-correlation interactions were treated using the generalized gradient approximation (GGA) within the Perdew-Burke-Ernzerhof (PBE) functional. Brillouin zone integration was carried out using a Monkhorst-Pack k-point mesh of 10x10x10. The plane-wave energy cutoff was set to 500 eV for expanding the wavefunctions. For the high temperature phase, which belongs to the P6/mmm space group, the optimized lattice constants were a = b = 5.467Å and c = 9.16Å. For the $\sqrt{3} \times \sqrt{3} \times 3$ charge density wave (CDW) phase, which belongs to the R32 space group, the calculated lattice constants were a = b = 9.46Å and c = 27.41Å. Spin-polarized calculations were performed to account for the Zeeman effect induced by the external magnetic field. The spin-up and spin-down components of the electronic structure were analyzed separately to extract the spin polarization at the Fermi level. An external magnetic field was introduced via the $B_{ext}$ (BEXT) parameter in VASP. The applied magnetic field induces Zeeman splitting of the electronic states, and the magnitude of the field was controlled by adjusting the value of BEXT (in eV).

## Data availability

Source data are provided with this paper. They have been deposited in a figshare repository with https://doi.org/10.6084/m9.figshare.29318846. The anisotropic Bragg peaks in Fig. 4a are plotted using data from ref. 9.

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

## Acknowledgements

This project was supported by NSF award DMR-2419425 awarded to J.X. and the Gordon and Betty Moore Foundation EPiQS Initiative, Grant # GBMF10276 awarded to J.X., W.R.M. and D.M. acknowledge support from the Gordon and Betty Moore Foundation's EPiQS Initiative, Grant GBMF9069 awarded to D.M. S.M. and R.P.M. acknowledge the support from AFOSR MURI (Novel Light-Matter Interactions in Topologically Non-Trivial Weyl Semimetal Structures and Systems), grant# FA9550-20–1-0322 awarded to D.M. J. L. and Y. Z. were supported by the National Science Foundation Materials Research Science and Engineering Center program through the UT Knoxville Center for Advanced Materials and Manufacturing (DMR-2309083) awarded to Y.Z.

## Author contributions

J.X. conceived and supervised the project. C.F., W.L., Y.W. and J.X. carried out the polarization rotation, Sagnac, reflectivity, specific heat and transport measurements. S.M., W.R.M., R.P.M. and D.M. grew the crystals. J.L. and Y.Z. performed the spin polarization calculations. J.X. drafted the paper with the input from all authors. All authors contributed to the discussion of the manuscript.

## Competing interests

The authors declare no competing interests.
