## [Transparent Peer Review file · Nature Communications]

Discovery of an Intermediate Nematic State in a Bilayer Kagome Metal ScV6Sn6

Corresponding Author: Professor Jing Xia

Version 0:

Reviewer comments:

Reviewer #1

(Remarks to the Author)

Farhang et al. present a combined transport, specific heat, optical reflectivity, and Sagnac interferometry study of ScV6Sn6. The focus of this work is a transition at $T = 86$ K, previously observed in reference 18, yet to be explained. Based on transport anisotropy, the lack of a time-reversal symmetry-breaking at $T = 86$ K, and the observed break of six-fold symmetry in between the T-CDW transition (93K) and $T = 86$ K, the authors propose a novel intermediate nematic phase in ScV6Sn6. In the discussion, the authors attribute the origin of this state to competition between charge density waves.

I find this work interesting and compelling, and I would recommend it for publication after the authors resolve some of my concerns.

1) Can the authors comment on the slight discrepancies between T^* star values between resistivity and optical measurements? While for TCDW the sharp decrease in resistivity is concomitant to the decrease of optical reflectivity, this doesn't seem to be the case at the second transition.

2) Additionally, while the resistivity monotonically decreases, the optical reflectivity and the optical depolarization show maximums in the intermediate region. Can the authors comment on this? Can they extract some information about the competition between the two orders?

3) Can the authors discuss their definition of 'local'? It seems they are using it to indicate that it is not due to macroscopic inhomogeneities, but local, which usually invokes intra-unit cell distances. Might I suggest intrinsic instead?

4) I would recommend that the authors try to move all discussion of Ref 8 in section 'Emergence of an intermediate phase inside the CDW state' to the introduction and highlight that the $T = 86$ K was observed but not discussed previously. In my first pass of the manuscript, I was left with the impression that this was an incremental work. The authors should emphasize the significance of their findings more strongly.

On a final note, and I recognize that this is not within the scope of this manuscript (I do not expect the authors to include this in a revised version of the manuscript), I encourage the authors to perform new high-resolution X-ray scattering measurements and IXS measurements in their high-quality samples.

Reviewer #2

(Remarks to the Author)

In this paper, Farhang et al. present detailed transport and optical measurements of the Kagome metal ScV6Sn6. They report a second phase transition at a temperature slightly below the previously known CDW transition temperature and interpret it as a nematic transition. They further argue that ScV6Sn6 offers a more accessible platform than the well-studied Kagome metal AV3Sb5. Although AV3Sb5 is one of the few Kagome-lattice superconductors and hence it may seem premature to claim that non-superconducting ScV6Sn6 is a better platform, exploring CDW and nematic phases in Kagome lattices remains an important challenge in condensed matter physics. Nevertheless, several key issues must be addressed before I recommend for publication.

Specifically, the evidence for a nematic phase between T_{CDW} and $T_{\text{*}}$ is suggestive but not yet conclusive.

1. The central experimental results are summarized in Figure 3, but the interpretation requires caution. Because the CDW in ScV_6Sn_6 has a three-dimensional stacking order, the bulk crystal may break rotational symmetry even if each layer remains symmetric. A similar stacking-driven symmetry breaking has been discussed in AV_3Sb_5 . Such an order is distinct from a true nematic state in which the lattice itself remains symmetric but the electronic system breaks rotational symmetry. Can the authors rule out a stacking-driven effect?

2. When a three-dimensional CDW forms, domains with different in-plane orientations often arise. In that case, the anisotropy in Figure 1a would be expected even without nematicity. Moreover, conductivity anisotropy frequently increases near phase transitions because of sample inhomogeneity. The authors note in line 156 that their crystals may exhibit some inhomogeneity. Could these factors account for the observed R_{xx}/R_{yy} deviation from unity?

3. If one assumes that the CDW sets in at T_{CDW} with a $\sqrt{3} \times \sqrt{3} \times 3$ superstructure and at $T_{\text{*}}$ with a $\sqrt{3} \times \sqrt{3} \times 2$ stacking (or vice versa), the data might be fully explainable without invoking nematic order. Have the authors considered and excluded this scenario or other competing CDW scenarios?

From the presented data, a transition at $T_{\text{*}}$ and an associated symmetry change seem clear. However, the claim that this change constitutes a nematic phase transition is not yet persuasive.

In line 86, please expand MOKE as “magneto-optic Kerr effect.”

Version 1:

Reviewer comments:

Reviewer #1

(Remarks to the Author)

The authors have satisfactorily addressed my concerns and those of referee 2. I recommend this manuscript for publication.

Reviewer #2

(Remarks to the Author)

I thank the authors for clarifying the reviewers' concerns. While they explain the difference between AV_3SB_5 and the present system in terms of stacking, including this explanation in the main text would benefit readers. Apart from this suggestion, I recommend the manuscript for publication in Nature Communications.

Dear referees:

Thank you for your professional assessments and insightful suggestions. Attached below are the summary of changes and point-to-point responses to comments and questions.

-Jing Xia, on behalf of all authors

Summary of Changes

Changes in the manuscript are highlighted in yellow.

1. The abstract is shortened to within the 150 words limit.
2. Section titles such as “RESULTS” and “DISCUSSIONS” are added.
3. “local” has been replaced by “intrinsic” when describing optical anisotropy.
4. The description of the second resistivity drop has been moved to the introduction line 45.
5. The following has been added to line 78: “We observe that T^* varies by ~ 1 K across different locations, while T_{CDW} remains uniform, suggesting that T^* marks a more delicate phase transition, more sensitive to local environments than the CDW transition”.
6. The following has been added to line 149: “To rule out sample inhomogeneity as the source of macroscopic transport anisotropy, we performed optical polarization rotation measurements.”.
7. The following has been added to line 217: “We note that none of the diffraction measurements have detected evidence of a long-range $\sqrt{3} \times \sqrt{3} \times 2$ CDW order, which could potentially break rotational symmetry.”.

Response to Referees

(Reponses are in blue for clarity.)

Reviewer #1 (Remarks to the Author):

Farhang et al. present a combined transport, specific heat, optical reflectivity, and Sagnac interferometry study of ScV6Sn6. The focus of this work is a transition at $T = 86$ K, previously observed in reference 18, yet to be explained. Based on transport anisotropy, the lack of a time-reversal symmetry-breaking at $T = 86$ K, and the observed break of six-fold symmetry in between the T-CDW transition (93K) and $T = 86$ K, the authors propose a novel intermediate nematic phase in ScV6Sn6. In the discussion, the authors attribute the origin of this state to competition between charge density waves.

I find this work interesting and compelling, and I would recommend it for publication after the authors resolve some of my concerns.

We thank the reviewer for his/her professional assessments and suggestions.

1) Can the authors comment in the slight discrepancies between T^* star values between resistivity and optical measurements? While for TCDW the sharp decrease in resistivity is concomitant to the decrease of optical reflectivity, this doesn't seem to be the case at the second transition.

Yes. T^* and T_{CDW} are determined using both bulk resistivity and local optical reflectivity. As shown in Extended Data Figure 1, copied below, T_{CDW} is consistent across all locations, marked by a sharp jump in reflectivity, and matches the resistivity data (as noted by Referee 1). In contrast, T^* varies by ~ 1 K between locations in optical reflectivity measurements and shows a location-dependent offset of ± 1 K from the bulk resistivity measurement. This supports our argument that T^* marks a more delicate phase transition driven

by competing CDW instabilities, making it more sensitive to local environments variations. In the manuscript, line 78, we have added the following sentence:

We observe that T^ varies by ~ 1 K across different locations, while T_{CDW} remains uniform, suggesting that T^* marks a more delicate phase transition, more sensitive to local environments than the CDW transition.*

Extended Data Figure 1. Temperature-dependent optical reflectivity ($1.55 \mu\text{m}$) measured at different locations.

2) Additionally, while the resistivity monotonically decreases, the optical reflectivity and the optical depolarization show maximums in the intermediate region. Can the authors comment on this? Can they extract some information about the competition between the two orders?

We thank Referee 1 for these insightful observations.

The peak in the optical depolarization signal, as noted in the manuscript line 163-175, indicates optical anisotropy, further corroborating the polarization rotation measurements.

The “maximum” in the optical reflectivity is more subtle. As shown in Extended Data Figure 1 copied above, the optical reflectivity generally decreases upon cooling, with a slightly curved plateau between T^* and T_{CDW} . This overall opposite temperature dependence of DC conductivity (inverse of resistivity) and optical reflectivity is explained in the manuscript (line 80): “In a metal that is isotropic in the ab -plane, a reduced optical reflectivity usually indicates a decreased plasma frequency and hence less numerous carriers, consistent with what was found in Hall effect measurements on a similar sample. The increased carrier mobility due to a decreased electronic scattering by CDW fluctuations accounts for the lower resistivity in the CDW phase, despite diminished density of states.” The small “maximum” in the

reflectivity between T^* and T_{CDW} suggests an unexpected deviation from the assumption of an in-plane isotropic metal. However, analyzing this subtle feature is nontrivial. Since nematicity has already been established locally through polarization rotation and Sagnac depolarization, we prefer to leave a deeper investigation of this subtle reflectivity feature to future work.

3) Can the authors discuss their definition of ‘local’? It seems they are using it to indicate that it is not due to macroscopic inhomogeneities, but local, which usually invokes intra-unit cell distances. Might I suggest intrinsic instead?

We thank referee 1 for highlighting the potential confusion between “intra-unit-cell” locality and the wavelength-scale locality that we intended to convey. We will adopt the term “intrinsic,” as suggested by the referee. The word “local” has been replaced with “intrinsic” in the revised manuscript line 77, 163, and figure captions.

4) I would recommend that the authors try to move all discussion of Ref 8 in section ‘Emergence of an intermediate phase inside the CDW state’ to the introduction and highlight that the $T = 86$ K was observed but not discussed previously. In my first pass of the manuscript, I was left with the impression that this was an incremental work. The authors should emphasize the significance of their findings more strongly.

We thank referee 1 for the suggestion. The following has been moved into the “Introduction” section:

Notably, below T_{CDW} , a pronounced second resistivity drop occurs at $T^ \approx 86$ K, as shown in Fig. 1a. This feature has been clearly observed only in ScV_6Sn_6 crystals with similar growth conditions, while hints of it appear as a long tail in temperature-dependent resistivity in other reports. The nature of this feature at T^* has not been previously discussed.*

On a final note, and I recognize that this is not within the scope of this manuscript (I do not expect the authors to include this in a revised version of the manuscript), I encourage the authors to perform new high-resolution X-ray scattering measurements and IXS measurements in their high-quality samples.

We thank referee 1 for the suggestion and are planning to conduct X-ray and IXS experiments to investigate this further.

Reviewer #2 (Remarks to the Author):

In this paper, Farhang et al. present detailed transport and optical measurements of the Kagome metal ScV_6Sn_6 . They report a second phase transition at a temperature slightly below the previously known CDW transition temperature and interpret it as a nematic transition. They further argue that ScV_6Sn_6 offers a more accessible platform than the well-studied Kagome metal AV_3Sb_5 . Although AV_3Sb_5 is one of the few Kagome-lattice superconductors and hence it may seem premature to claim that non-superconducting ScV_6Sn_6 is a better platform, exploring CDW and nematic phases in Kagome lattices remains an important challenge in condensed matter physics. Nevertheless, several key issues must be addressed before I recommend for publication.

We thank referee 2 for the thoughtful assessments and suggestions. We agree that it may be premature to claim that the non-superconducting ScV_6Sn_6 is a better platform than AV_3Sb_5 . Accordingly, we have removed such statements, including changing the final sentence of the abstract to:

Our findings highlight ScV_6Sn_6 as a prime candidate for realizing symmetry-breaking phases driven by CDW competition, kagome physics, and Van Hove singularities.

Specifically, the evidence for a nematic phase between T_{CDW} and T_{N}^* is suggestive but not yet conclusive.

1. The central experimental results are summarized in Figure 3, but the interpretation requires caution. Because the CDW in ScV_6Sn_6 has a three-dimensional stacking order, the bulk crystal may break rotational symmetry even if each layer remains symmetric. A similar stacking-driven symmetry breaking has been discussed in AV_3Sb_5 . Such an order is distinct from a true nematic state in which the lattice itself remains symmetric but the electronic system breaks rotational symmetry. Can the authors rule out a stacking-driven effect?

Referee #2 raises the possibility that the observed rotational symmetry breaking may arise from the 3-dimensional CDW stacking order. While we did mention in the manuscript a potential $2\pi/3$ phase shift in ScV_6Sn_6 , analogous to the proposed π phase shift between CDW layers as the origin of symmetry breaking in AV_3Sb_5 , we believe this scenario is unlikely to account for the symmetry breaking observed in our measurements. This stacking configuration would reduce the in-plane rotational symmetry only locally, from 6-fold to 3-fold. However, over the many layers the stacking sequence effectively preserves six-fold symmetry on average. This contrasts with the π phase shift stacking proposed in the $2\times 2\times 2$ CDW ordered phase AV_3Sb_5 , which can result in a reduction to 2-fold symmetry throughout the entire bulk of the crystal, and symmetry cannot be effectively restored through averaging over layers. In our case, a $2\pi/3$ interlayer phase shift would not result in a global reduction of in-plane symmetry and therefore is unlikely to be the origin of the rotational symmetry breaking observed.

2. When a three-dimensional CDW forms, domains with different in-plane orientations often arise. In that case, the anisotropy in Figure 1a would be expected even without nematicity. Moreover, conductivity anisotropy frequently increases near phase transitions because of sample inhomogeneity. The authors note in line 156 that their crystals may exhibit some inhomogeneity. Could these factors account for the observed R_{xx}/R_{yy} deviation from unity?

As referee 2 rightly pointed out, macroscopic transport anisotropy could potentially result from sample inhomogeneity. This is precisely why we conducted local polarization rotation and Sagnac depolarization measurements, which are not susceptible to such artifacts and provide more direct evidence of nematicity. We recognize that this rationale may not have been clearly stated in the manuscript. To address this, we have added the following clarification at line 149:

“To rule out sample inhomogeneity as the source of macroscopic transport anisotropy, we performed optical polarization rotation measurements, which also serve to determine the anisotropic principal axes.”

3. If one assumes that the CDW sets in at T_{CDW} with a $\sqrt{3} \times \sqrt{3} \times 3$ superstructure and at T_{N}^* with a $\sqrt{3} \times \sqrt{3} \times 2$ stacking (or vice versa), the data might be fully explainable without invoking nematic order. Have the authors considered and excluded this scenario or other competing CDW scenarios?

Referee 2 is correct that an intermediate $\sqrt{3} \times \sqrt{3} \times 2$ ordered CDW phase might lead to rotational symmetry breaking in the narrow temperature range. However, none of the numerous diffraction measurements (Nat Commun 14, 6646 (2023), Nat Commun 14, 7671 (2023), Phys. Rev. Mater. 7, 104201 (2023) to name a few) have seen evidence of such long-range $\sqrt{3} \times \sqrt{3} \times 2$ order, while a short-range order won't yield any observable anisotropy in our transport or optical experiments. To clarify this point, we have added the following sentence to line 217:

We note that none of the diffraction measurements^{20,27,47} have detected evidence of a long-range $\sqrt{3} \times \sqrt{3} \times 2$ CDW order, which could potentially break rotational symmetry.

From the presented data, a transition at T* and an associated symmetry change seem clear. However, the claim that this change constitutes a nematic phase transition is not yet persuasive.

We thank Referee 2 for recognizing that our data clearly demonstrate a symmetry change at T*, but we respectfully disagree with the assertion that this does not constitute a nematic phase. While it might be tempting to define a “nematic phase” as purely electronic, without involving lattice symmetry breaking, in practice, electronic and lattice nematicity are inherently intertwined and cannot be cleanly separated. For example, in the case of AV₃Sb₅, some studies emphasize the electronic nematic aspect, while others focus on the lattice contribution:

“...Electronic nematicity, in which rotational symmetry is spontaneously broken by electronic degrees of freedom...” *Nature* 604, 59–64 (2022).

“...our measurement is most likely due to the π phase shift of the stacking between CDW layers (Fig. 1b and Extended Data Fig. 1a), because the onset temperature of birefringence coincides with TCDW in all three compounds...” *Nat. Phys.* 1470–1475 (2022).

We note that the intermediate state observed in this work emerges spontaneously in conjunction with the CDW transition. It arises from electronic and/or lattice nematicity and is plausibly a result of competing CDW instabilities.

In line 86, please expand MOKE as “magneto-optic Kerr effect.”

We have expanded MOKE as “magneto-optic Kerr effect (MOKE)”.

Response to Referees

-Jing Xia

(Reponses are in blue for clarity.)

Reviewer #1 (Remarks to the Author):

The authors have satisfactorily addressed my concerns and those of referee 2. I recommend this manuscript for publication.

We thank the reviewer #1 for his/her valuable time and thoughtful evaluation of our manuscript, and the recognition of our work.

Reviewer #2 (Remarks to the Author):

I thank the authors for clarifying the reviewers' concerns. While they explain the difference between AV₃Sb₅ and the present system in terms of stacking, including this explanation in the main text would benefit readers. Apart from this suggestion, I recommend the manuscript for publication in Nature Communications.

We thank the reviewer #2 for his/her valuable time and thoughtful evaluation of our manuscript, and the recognition of our work. Following reviewer #2's suggestion, we have added a shortened version of our clarification in the previous response letter to the main text line 240:

“This stacking would locally reduce rotational symmetry from six-fold to three-fold, but preserve six-fold symmetry on average across layers. In contrast, the π phase shift in the proposed $2 \times 2 \times 2$ CDW phase of AV₃Sb₅ break symmetry globally to two-fold, not being restored by averaging. Thus a $2\pi/3$ phase shift is unlikely to account for the rotational symmetry breaking we observe.”